# Flywheel Romanian Deadlift: Intra- and Inter-Day Kinetic and Kinematic Reliability of Four Inertial Loads Using Cluster Sets

**DOI:** 10.3390/jfmk9010001

**Published:** 2023-12-19

**Authors:** Shane Ryan, Rodrigo Ramirez-Campillo, Declan Browne, Jeremy Moody, Paul J. Byrne

**Affiliations:** 1Department of Health and Sport Sciences, South East Technological University (Kilkenny Road Campus), R93 V960 Carlow, Ireland; shane.ryan@postgrad.setu.ie (S.R.); declan.browne@setu.ie (D.B.); 2Exercise and Rehabilitation Sciences Institute, School of Physical Therapy, Faculty of Rehabilitation Sciences, Universidad Andres Bello, Santiago 7591538, Chile; rodrigo.ramirez@unab.cl; 3Cardiff School of Sport and Health Sciences, Cardiff Metropolitan University, Cardiff CF5 2YB, UK; jmoody@cardiffmet.ac.uk

**Keywords:** eccentric overload, isoinertial, power, resistance training, flywheel, attentional focus

## Abstract

The primary aim of this study was to investigate the intra- and inter-day reliability of flywheel cluster set training in concentric power (CON), eccentric power (ECC), and ECC overload during the Romanian deadlift exercise (RDL). A secondary aim was to assess the acute effect of internal and external attentional focus instructions on mean power when performing the flywheel RDL. Fourteen collegiate male field sport athletes (age, 23.3 ± 3.7 years; mass, 80.8 ± 9.9 kg; height, 1.79 ± 0.06 m) were randomized into internal (*n* = 7) or external (*n* = 7) attentional focus groups and attended four testing sessions, with a between-session separation of 7 days. Sessions consisted of four cluster sets of fifteen repetitions “excluding momentum repetitions” (4 × (5 + 5 + 5)) using a specific inertial load (0.025, 0.050, 0.075, and 0.100 kg·m^−2^) for a given set in a randomized ascending or descending order. Cluster sets were separated by a 45 s intra-set rest period. Both instructional focus groups attained familiarization, although the time taken to achieve familiarization (outcome stability) differed between groups. The external instructional group attained familiarization post-session 2 (Cohen’s *d* (ES), ES = 0.11–0.65) with little volatility between performance measures (CV% = 4.61–9.59). Additionally, the internal group reported inconsistencies among all inertial loads, reporting large differences in MP in the 0.100 kg·m^−2^ inertial load from day 2 to day 3 (ES = 1.22) and both 0.050 kg·m^2^ (*p* = 0.010) and 0.075 kg·m^−2^ (*p* = 0.016) between day 3 and day 4. The flywheel RDL cluster set approach is a reliable training modality for maintaining mean power output during cluster set repetitions.

## 1. Introduction

Flywheel iso-inertial training (FIT) has grown exponentially in recent decades given its utilization of principles of inertia and eccentric (ECC) overload capabilities [1]. Moreover, the portability of FIT devices used to apply these principles is an advantage for athlete physical conditioning. Initially designed by NASA to mitigate muscle atrophy resulting from prolonged exposure to the anti-gravitational effects of space travel [2,3], like traditional weight-bearing methods, the innovative FIT model can be broken into two phases. Firstly, the concentric (CON) phase, whereby the acceleration of the flywheel disc occurs via tension in the tether connecting the user to a weighted flywheel disc. The CON phase is followed immediately by a second phase, also known as the “breaking phase” [4], during which, the user is responsible for generating ECC force to decelerate the weighted flywheel [5]. It should be noted that, for the ECC overload stimulus to be attained, the user must delay the breaking action until the final third of the ECC movement [6]. Thereafter, inertial force adjustments can be made relating to the geometric (diameter and thickness) properties of the disc(s) [7]. However, there is a scarcity of literature investigating the familiarization constraints, reliability, and efficiency of FIT, particularly for exercises such as the Romanian deadlift.

A familiarization period, initial momentum repetitions, and adequate progressions are required for stability and optimal FIT performance [5,8]. To date, two studies have examined the familiarization period during a quarter-squat exercise and the stability of performance measures with both CON and ECC power output (mean and peak), showing good-to-excellent levels of reliability when using the exercise [7,9]. During one of these studies, attentional focus was examined as a mode to enhance the skill acquisition process and decrease the required number of sessions to reach familiarization [9]. The findings suggested the superiority of external instructional instructions during this process. However, to date, no research has examined this approach using an RDL exercise. Furthermore, although there have been rapid increases in the body of literature surrounding FIT, there is still uncertainty regarding training guidelines outside of the quarter-squat exercise [4]. This could be due to the technical components involved in performing various hip extension exercises [10].

Hip extension exercises are commonly utilized by practitioners to strengthen the posterior chain, although practical applications within the FIT literature are conspicuous in their absence [4,11]. Hip extension exercise is associated with the increased acceleration of the body’s vertical and horizontal force production, enabling athletes to increase movements such as change of direction (COD), jumping, and sprinting [12]. The Romanian deadlift (RDL) exercise is a large multi-joint movement often incorporated to enhance hip extension and flexion [13]. O’Brien et al. [14] reported the biomechanical disadvantages of the RDL during the FIT exercise, concluding that the postural angle of the “supine open chain” during the loading phase influences the inertial power relationship. Moreover, posterior chain injuries are among the most commonly reported within sports medicine for elite athletes [15,16]. A study by De Keijzer et al. was the first FIT study to address the inter-session reliability of two hip extension exercises (RDL and leg curl) [17]. This study reported conflicting observations when comparing the inter-session reliability of the two exercises [17]. The RDL exercise proved good to excellent for both CON (ICC = 0.85–0.97) and ECC (ICC = 0.88–0.98) peak power measures. However, the leg curl exercise reported acceptable-to-good levels of reliability with inconsistent performance measures between sessions.

A recent study reported the effects of a FIT cluster set (CS) training approach utilizing intra-set rest periods during the familiarization process to reinforce technical form instructions and prevent fatigue accumulation during the familiarization process [9]. The study established excellent reliability in the CS approach, independent of inertial load. However, familiarization was obtained significantly earlier with external coaching instructions for the 0.050 kg·m^−2^ and 0.075 kg·m^−2^ inertial loads. To the best of the authors’ knowledge, this was the only FIT study to incorporate specific coaching instructions. Moreover, previous coaching instructions within the FIT literature are limited to reports where subjects “perform the CON phase as fast as possible” and “delay the breaking phase until the final third” [18,19]. A meta-analysis reported four sets of seven repetitions as the most common prescription for FIT [17]. However, conflicting reports by Sabido et al. [7] found significant decrements in peak and mean power outputs in sets exceeding six continuous repetitions, independent of inertial load (e.g., repetition 5 using a 0.05 kg·m^−2^ load). Thus, the CS approach may prove beneficial for decreasing the high levels of fatigue associated with ECC exercise [20]. Furthermore, the lack of practical guidelines for hip extension exercises within the FIT literature should be rectified.

To the best of our knowledge, only one previous study has investigated a familiarization phase using a flywheel CS training approach [9]. Thus, the primary aim of this study was to investigate the intra- and inter-day reliability of flywheel cluster set training in concentric power (CON), eccentric power (ECC), and ECC overload during the RDL. A secondary aim was to assess the acute effect of internal and external attentional focus instructions on mean power when performing the flywheel RDL.

## 2. Materials and Methods

### 2.1. Experimental Approach to the Problem

This study used a randomized repeated measures design, whereby participants were randomly allocated into one of two instructional focus groups, (1) internal and (2) external, to assess the acute effects of verbal attentional focus instructions on kinetic and kinematic measures when performing a cluster set. Participants underwent a repeated measures design where male field sport athletes completed four FIT testing sessions, separated by 7 days to determine both intra and inter-day reliability peak/mean power output and ECC overload during an RDL (see Figure 1).

### 2.2. Participants

A total of 20 amateur male collegiate field sport athletes (Gaelic football, hurling, and soccer) voluntarily participated in this study (age, 23.3 ± 3.7 years; mass, 80.8 ± 9.9 kg; height, 1.79 ± 0.06 m). Inclusion criteria dictated that participants be deemed “resistance-trained individuals” based on training experience (2.0 ± years) and a relative one-repetition maximum (1RM) lower-body-strength level (1.5 × body weight) for the RDL [21]. Participants were in the off-season of their chosen sport, which included reduced training schedules. On average, participants were training three times per week (one field session, two weight training). Participants were encouraged to continue their normal training while testing commenced and not consume any type of stimulant for the 48 h prior to testing sessions. At the beginning of testing, all participants were required to have no previous flywheel training experience, potentially influencing their ability during the familiarization process. Failure to meet these criteria meant participants were excluded from participation. Thus, for reasons outside of the researchers’ control, 6 participants were excluded from the data analysis. A GPower sample size calculation was used prior to testing (two-way repeated measures ANOVA) considering an effect size of 0.6 [22], an alpha error of <0.05, a non-sphericity correction of € = 1, a correlation between the repeated measures of 0.5, and a desired power (1 − β error) of 0.95; the total sample size resulted in 20 participants. Experimental procedures were completed following the Declaration of Helsinki and approved by the university research ethics board (Ethics Board Application Number: 281).

### 2.3. Procedures

All flywheel testing sessions commenced with a warm-up consisting of a 5 min self-paced jog that was followed by five dynamic stretches (quadriceps, hamstrings, gluteal, adductors, and gastrocnemius), where all stretches were performed over a 10 m distance for a total of 14 repetitions per leg [23]. Subsequently, all participants performed a submaximal warm-up set of 10 repetitions using a 0.05 kg·m^−2^ inertial load. The FIT cluster protocol employed consisted of 4 sets of 15 (4 × (5 + 5 + 5)) repetitions of an RDL exercise performed on a flywheel device (kBox 4, Exxentric, AB TM, Bromma, Sweden). Each complete cluster block (CB) was separated by a 45 s intra-set rest period [9,24]. The CBs included two “momentum repetitions”, whereby the first and second repetitions of each CB were used to increase the velocity of the weighted disc and were excluded from the data analysis. A rest period of 4 min was provided between completed sets (15 repetitions). The order of the inertial loads was randomized in terms of being performed in an ascending order from 0.025 kg·m^2^ to 0.100 kg·m^2^ or in a descending order from 0.100 kg·m^2^ to 0.025 kg·m^2^ [7,9]. Participants attended four testing sessions separated by 7 days.

Range of motion was standardized from an upright position to the apex of the patella, which was individualized for each participant using tape. This end range was used to target the posterior chain (hamstring and gluteal) in an extended position. All participants began the movement in an upright position with the weighted load located in a frontal position. During the ECC phase of the movement, the participant underwent hip flexion while loading the posterior chain in the lengthened position. Moreover, once depth at the lower aspect of the patella was obtained, the participant extended their hips forward and began ascending to the upright position they began with, maintaining a neutral posture throughout [12,25]. During the CON phase of the exercise, participants were instructed to perform the movement as fast as possible with maximal effort, followed by a delay in the breaking action until the final third of the ECC phase [20]. All participants were requested to use weightlifting straps to prevent grip strength from being a limiting factor.

During all testing sessions, participants in the attentional focus groups received either internal or external focus instructions to aid in the movement. The instructional focus instructions were targeted at the individuals’ abilities to create flexion and extension at the hip joint while allowing for power production throughout the exercise. These instructions were developed by the lead author. During the FIT cluster RDL, the internal focus instruction group received the following instructions: “Drive your hips forward” and “Break and absorb at your hips”. The external focus instruction group received the following instructions: “Drag the bar towards the hips” and “Control the bar downwards while loading”. All coaching instructions were provided prior to the exercise and then reiterated during each 45 s intra-set rest period to aid in the motor learning process.

The kMeter application (app) was used to record mean and peak power scores to assess performance decrements. Moreover, the CON/ECC ratio was also assessed for each load. The kMeter app has been found to be a reliable measure of mean CON and ECC power outputs during FIT [26]. Familiarization was achieved when a participant performed two sessions in succession without a significant difference in mean force output [7]. The ECC overload was calculated in both absolute (Nm = ECC peak force/CON peak force) and relative values (ECC peak force/100/CON peak force/100).

### 2.4. Statistical Analysis

Data normality was verified with the Shapiro–Wilks test, and means ± standard deviations (SDs) were calculated for all measures. Intra- and inter-day reliability was assessed using an intra-class correlation coefficient (3.1, 2-way mixed model with consistency and average effect measure) with 95% confidence intervals. Single measures from this ICC model and Cronbach’s alpha were also reported for individual and mean power output during cluster sets. The interpretations of ICC values were poor (0.00–0.49), moderate (0.50–0.69), high (0.70–0.89), or very high (≥0.9) [27]. Absolute reliability was assessed using coefficient of variance (CV%) and standard error of measurement (SEM). The CV% was calculated as follows: standard deviation/mean *100 with a cut-off point for acceptability set at 10% [28]. The SEM was calculated as follows: SEM = SD × √ (1 − ICC) [29]. When significance was reported between interactions, a post hoc analysis was conducted using a Bonferroni correction. Statistical significance was set at *p* ≤ 0.05. Cohen’s *d* effect size (ES) was calculated and interpreted as trivial (<0.2), small (0.2–0.5), moderate (>0.5–0.8), or large ((>0.8) [30]. The ES was used to estimate where in the familiarization process the greatest learning effect occurred between subsequent sessions (day 1 vs. day 2, day 2 vs. day 3, day 3 vs. day 4). The smallest worthwhile change (SWC) was calculated to assess meaningful changes between measures at 0.5* the standard deviation for each CS. Typical error (TE) was used to assess the smallest worthwhile change and interpreted as marginal (TE < SWC) or good (TE > SWC) [31]. Statistical analyses were conducted using IBM SPSS Version 27 (SPSS, Inc., Chicago, IL, USA).

## 3. Results

The mean power output and standard deviations of each individual inertial load across separate testing days are displayed for both instructional groups in Table 1. Intra- and inter-day reliability scores for MP output using each inertial load (0.025, 0.050, 0.075, and 0.100 kg·m^−2^) are displayed in Table 2 and Table 3, respectively. Intra-day reliability for the external group showed excellent reliability (α *=* 0.93 and 0.99) independent of the inertial load used. However, the internal group recorded inconsistencies throughout the testing days and inertial loads, reporting moderate-to-excellent reliability (α *=* 0.67–0.99). Furthermore, the inter-day reliability of MP output is displayed in Figure 2. External familiarization was deemed to have been achieved by session 3. Significant improvements were reported in MP output from session 2 to session 3, independent of inertial load (0.025 kg·m^2^ (*p =* 0.001), 0.050 kg·m^2^ (*p* = 0.002), 0.075 kg·m^2^ (*p =* 0.002), and 0.100 kg·m^2^ (*p =* 0.002)). Thereafter, stability was reported in MP output across all inertial loads (*p =* 0.069–0.818). Moreover, the internal group showed delayed stability in MP, and both 0.050 kg·m^2^ (*p =* 0.010) and 0.075 kg·m^2^ (*p =* 0.016) showed a significant increase from sessions 3 to 4.

Variation in external MP showed excellent reliability (CV%) in the external group post-session 1, independent of inertial load (CV% *=* 4.61–9.59). The internal group showed an excellent CV% throughout all sessions using the 0.075 kg·m^−^^2^ and 0.100 kg·m^−^^2^ inertial loads (CV% *=* 6.45–8.70). Thereafter, the internal group showed delayed levels of variance in the 0.025 kg·m^−^^2^ (session 3 CV% *=* 10.09) and 0.050 kg·m^−^^2^ (session 2 CV% *=* 14.45) inertial loads. Furthermore, ES showed trivial changes in performance between testing days 1 and 2 in the internal group (ES *=* 0.07–0.21). In addition, inconsistencies were observed dependent on inertial load (see Table 4). Moreover, a large ES (ES *=* 1.22) was reported in the 0.100 kg·m^−^^2^ inertial load, corresponding to the timing of familiarization in the internal group. Similarly, the external instructional group reported moderate ES between sessions 1, 2, and 3, demonstrating increases in performance as participants became accustomed to the exercise (ES *=* 0.11–0.65). Thereafter, trivial changes were observed post-session 3 (ES *=* 0.20).

Table 5 and Table 6 display the influence of inertial load on CON/ECC power output. The ECC overload ratio is displayed for both instructional groups. Independent of the instructional group, CON power was greater with the 0.025 kg·m^−^^2^ inertial load. However, the 0.050 kg·m^−^^2^ inertial load resulted in similar levels during day 4 MP output measures. Moreover, the 0.050 kg·m^−^^2^ inertial load produced greater CON output than the 0.075 kg·m^−^^2^ inertial load, and the 0.075 kg·m^−^^2^ load resulted in substantially greater CON output than the 0.100 kg·m^−^^2^ inertial load. Contrary to the FIT literature, the 0.025 kg·m^−^^2^ inertial load showed the greatest ECC output during session 1 in comparison with all other inertial loads. During this initial stage, it was theorized that skill acquisition was more difficult to attain using heavier inertial loads. Moreover, post-session 1, there was a noticeable difference between the instructional groups, as the internal instructional group showed significantly lower ECC overload using the 0.100 kg·m^−^^2^ inertial load during sessions 3 and 4 (see Table 5 and Table 6).

## 4. Discussion

The primary aim of the current study was to investigate the intra- and inter-day reliability of flywheel cluster set training in concentric power (CON), eccentric power (ECC), and ECC overload during the RDL. A secondary aim was to assess the acute effect of internal and external attentional focus instructions on mean power when performing the flywheel RDL. The findings of the present study align with similar FIT research, in that a minimum of 2–3 sessions are required during the familiarization process [7]. Moreover, the present findings suggest the specificity of coaching instructions plays a significant role in the skill acquisition process and the stability of both CON and ECC power output (Figure 2). Furthermore, there were no significant differences in MP output between groups across the four testing sessions. However, significant (*p* ≤ 0.05) results were noted regarding (i) the time taken to reach stability (external instructions = three sessions; internal instructions = four sessions) and (ii) the reliability of both intra- and inter-day performance metrics (peak/mean power output) independent of inertial load. Thus, the group that received external coaching instructions required three sessions to reach stability in power output; however, the group that received internal coaching instructions required four sessions. When working with athletes, time can be limited for physical conditioning, and requiring fewer sessions is beneficial from a time management point of view.

To date, FIT research has reported excellent levels of reliability when it comes to the quarter-squat exercise [5,9]. Conversely, the RDL exercise has seen little in-depth research examining the reliability of its performance metrics [12,14]. Interestingly, this study reports different reliability variables (ICC and CV%) dependent on specific instructional coaching instructions (internal CV% = 5.58–14.45% and ICC = 0.67–0.99; external = CV% = 4.61–11.80% and ICC = 0.93–0.99). Moreover, both ICC and CV% values increased in strength post-session 1, coinciding with previous FIT literature suggesting that 2–3 familiarization sessions are required [7]. Furthermore, inter-day reliability showed significant changes in performance between the instructional groups (Table 3 and Table 4). The external group showed a significant change in performance between sessions 2 and 3 (*p* = 0.002), followed by a plateau in MP output post-session 3, independent of inertial load (*p* = 0.69–0.818). In contrast, at the same timepoint, the internal instructional group showed inconsistencies throughout the inertial loads. The 0.025 kg·m^−2^ load experienced no fluctuations in MP output throughout all testing sessions, while the internal group experienced delayed skill acquisition using the 0.050 kg·m^−2^ (*p* = 0.010) and 0.075 kg·m^−2^ (*p* = 0.016) inertial loads. Moreover, the heaviest inertial load (0.100 kg·m^−2^) reported similar trends to that of the external group, encountering a significant improvement in performance between sessions 2 and 3 (*p* = 0.001), followed immediately by a plateau in performance between sessions 3 and 4 (*p* = 0.14).

To ensure that FIT has the appropriate training impact, flywheel kinematic and kinetic data must be stabilized during a familiarization period [32]. Moreover, the FIT literature has frequently reported decrements in both mean and peak power production during sets of ≥6 repetitions because of fatigue accumulation and the stability of flywheel velocity [5,7]. Comparatively, a recent study suggested the cluster approach as a viable and highly reliable method for maintaining power output during FIT [9]. The current study reinforces the CS approach, optimizing the familiarization process and reducing inter-day variance throughout the familiarization period [5,6]. Furthermore, the external group experienced the greatest variance in CV% during session 1 (CV% = 6.49–11.80%), thus aligning with Beato et al. [5], in that excellent levels of reliability could be met over two sessions. It should be mentioned that the best CB throughout the testing days differed between the three sets (five repetitions), although there were no discernible variations between the sets (CB1, CB2, and CB3). An analysis of intra-set repeats for both CON and ECC production independent of inertial load was conducted to look for output reductions brought on by CS. No power decreases were noticed during any of the internal cluster sets according to the ES data.

Sporting success has widely been linked to an athlete’s capacity to produce high muscular power and the velocity at which these movements can be performed in a controlled manner [24,33,34]. Thus, with FIT being so dependent on CON output and velocities, it seems reasonable to desire low levels of fatigue during training. It is encouraging to contrast the results of the current study with Tufano et al. [35], who reported no significant difference in mean and peak power output while comparing traditional and cluster training methods during a back-squat. Moreover, in a context similar to the FIT familiarization period needed to attain stability in performance metrics, a study by Ritti-Dias et al. [36] reported similar familiarization periods are required for untrained individuals to achieve stability in 1RM during a back-squat. As most flywheel research is conducted on participants unfamiliar with the training modality, it seems apparent that a similar timeframe is necessary.

This study encountered a few limitations, the primary limitation being the limited concluding sample size (*n* = 14). Resource constraints were the determining factor behind this low sample number because of restrictions in place due to the COVID-19 pandemic [37]. Moreover, COVID-19 presented an obstacle when recruiting sufficient participants, aided by high drop-out rates. Furthermore, as per recommendations by Ryan et al. [9], a potential limitation of this cluster approach is the quantity of rest time (45 s per cluster block), which may not be widely accepted through training logistics. This is a dilemma in fatigue manipulation and optimal loading parameters when training for maximal power production.

## 5. Conclusions

In conclusion, this is the first study to examine the inter- and intra-day reliability of a FIT cluster protocol during an RDL exercise. It is interesting to note the positive applications of cluster sets as a fatigue manipulation strategy and the associated excellent levels of reliability (ICC and CV%). Our findings suggest that the inclusion of a cluster block may be an effective training mode to incorporate FIT strategies into, especially when peak CON and ECC power output are the desired training goals. Furthermore, it should be noted that the precision of attentional focus instructions plays a significant role in the skill acquisition process. Therefore, it is essential for coaches and sports scientists alike to utilize external coaching instructions where possible to sufficiently achieve the ECC overload training goal of flywheel training.

## Figures and Tables

**Figure 1 jfmk-09-00001-f001:**
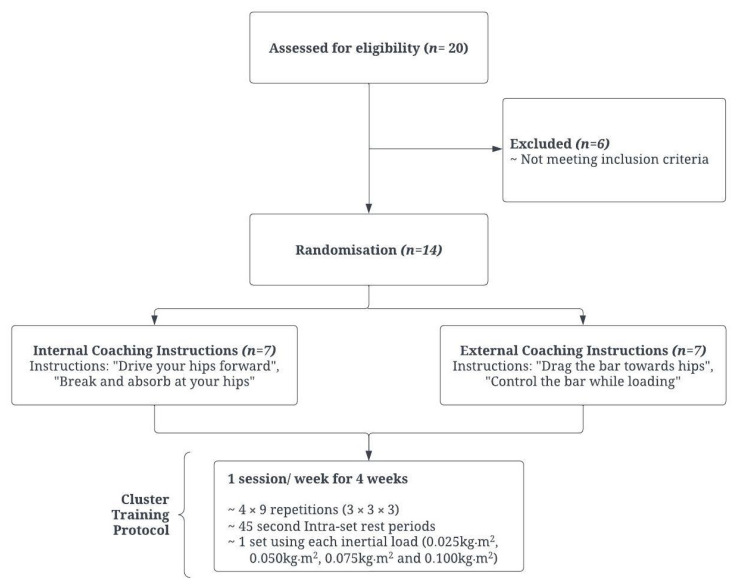
Testing procedure, including specific coaching instructions and weekly flywheel training program.

**Figure 2 jfmk-09-00001-f002:**
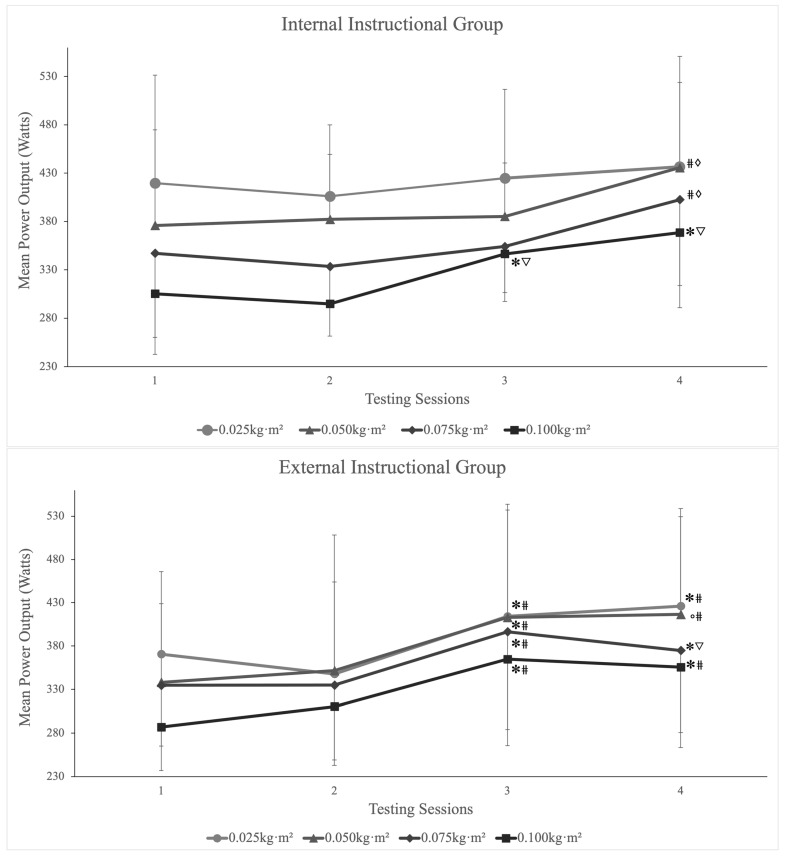
Inter-day inter-set mean power (MP) by inertial load during the flywheel Romanian deadlift exercise from sessions 1 to 4. * = significantly greater than day 1 (*p* ≤ 0.05); ° = significantly greater than day 1 (*p* ≤ 0.001); # = significantly greater than day 2 (*p* ≤ 0.05); ∇ = significantly greater than day 2 (*p* ≤ 0.001); ◊ = significantly greater than day 3 (*p* ≤ 0.05).

**Table 1 jfmk-09-00001-t001:** Mean power output (W; mean ± standard deviation) by inertial load and familiarization testing day for the internal and external attentional focus groups.

Internal Group				
Inertial Loads	Day 1	Day 2	Day 3	Day 4
0.025 kg·m^2^	420 ± 112	406 ± 74	425 ± 92	437 ± 114
0.050 kg·m^2^	376 ± 99	382 ± 67	385 ± 55	436 ± 88
0.075 kg·m^2^	347 ± 87	333 ± 40	354 ± 48	403 ± 89
0.100 kg·m^2^	305 ± 63	295 ± 33	346 ± 49	368 ± 78
**External group**				
Inertial loads	Day 1	Day 2	Day 3	Day 4
0.025 kg·m^2^	370 ± 96	348 ± 106	414 ± 123	426 ± 103
0.050 kg·m^2^	338 ± 91	352 ± 157	413 ± 131	417 ± 122
0.075 kg·m^2^	335 ± 69	358 ± 86	396 ± 112	375 ± 94
0.100 kg·m^2^	287 ± 50	310 ± 67	365 ± 99	356 ± 92

**Table 2 jfmk-09-00001-t002:** Intra-day reliability statistics reporting Cronbach’s α for both ICC single and average measures and 95% confidence intervals (CI) for 0.025, 0.050, 0.075, and 0.100 kg·m^2^ flywheel Romanian deadlift mean power outputs between complete sets over 4 familiarization testing sessions for the internal and external attentional focus groups.

Internal Group					
Inertial Load	Cronbach’s α	Single Measure	95% CI	Average Measure	95% CI
0.025 kg·m^2^ (Day 1)	0.97	0.93	0.78–0.99	0.98	0.92–1.00
0.025 kg·m^2^ (Day 2)	0.94	0.79	0.44–0.96	0.92	0.70–0.98
0.025 kg·m^2^ (Day 3)	0.97	0.83	0.34–0.97	0.94	0.61–0.99
0.025 kg·m^2^ (Day 4)	0.99	0.96	0.87–0.99	0.99	0.95–1.00
0.050 kg·m^2^ (Day 1)	0.98	0.95	0.82–0.99	0.98	0.93–1.00
0.050 kg·m^2^ (Day 2)	0.87	0.72	0.29–0.94	0.88	0.54–0.98
0.050 kg·m^2^ (Day 3)	0.88	0.72	0.32–0.94	0.89	0.59–0.98
0.050 kg·m^2^ (Day 4)	0.95	0.86	0.60–0.97	0.95	0.82–0.99
0.075 kg·m^2^ (Day 1)	0.91	0.78	0.42–0.95	0.92	0.68–0.98
0.075 kg·m^2^ (Day 2)	0.83	0.65	0.17–0.92	0.85	0.39–0.97
0.075 kg·m^2^ (Day 3)	0.67	0.35	−0.38–0.80	0.62	−0.13–0.92
0.075 kg·m^2^ (Day 4)	0.95	0.87	0.62–0.97	0.95	0.83–0.99
0.100 kg·m^2^ (Day 1)	0.86	0.68	0.27–0.93	0.87	0.52–0.98
0.100 kg·m^2^ (Day 2)	0.71	0.48	−0.30–0.87	0.73	−0.96–0.95
0.100 kg·m^2^ (Day 3)	0.95	0.88	0.64–0.98	0.96	0.84–0.99
0.100 kg·m^2^ (Day 4)	0.97	0.92	0.75–0.98	0.97	0.90–1.00
**External Group**					
Inertial load	Cronbach’s α	Single Measure	95% CI	Average Measure	95% CI
0.025 kg·m^2^ (Day 1)	0.97	0.92	0.74–0.99	0.97	0.89–1.00
0.025 kg·m^2^ (Day 2)	0.96	0.89	0.67–0.98	0.96	0.85–0.99
0.025 kg·m^2^ (Day 3)	0.98	0.95	0.85–0.99	0.98	0.94–1.00
0.025 kg·m^2^ (Day 4)	0.99	0.97	0.88–0.99	0.99	0.96–1.00
0.050 kg·m^2^ (Day 1)	0.96	0.90	0.69–0.98	0.96	0.87–0.99
0.050 kg·m^2^ (Day 2)	0.96	0.89	0.67–0.98	0.96	0.86–0.99
0.050 kg·m^2^ (Day 3)	0.96	0.89	0.68–0.98	0.96	0.87–0.99
0.050 kg·m^2^ (Day 4)	0.98	0.94	0.81–0.99	0.98	0.93–1.00
0.075 kg·m^2^ (Day 1)	0.96	0.89	0.67–0.98	0.96	0.86–0.99
0.075 kg·m^2^ (Day 2)	0.97	0.93	0.75–0.99	0.97	0.90–1.00
0.075 kg·m^2^ (Day 3)	0.98	0.95	0.83–0.99	0.98	0.94–1.00
0.075 kg·m^2^ (Day 4)	0.95	0.88	0.64–0.98	0.96	0.84–0.99
0.100 kg·m^2^ (Day 1)	0.93	0.81	0.49–0.96	0.93	0.74–0.99
0.100 kg·m^2^ (Day 2)	0.95	0.85	0.59–0.97	0.95	0.81–0.99
0.100 kg·m^2^ (Day 3)	0.98	0.96	0.85–0.99	0.98	0.94–1.00
0.100 kg·m^2^ (Day 4)	0.95	0.86	0.60–0.97	0.95	0.82–0.99

**Table 3 jfmk-09-00001-t003:** Inter-day reliability statistics reporting Cronbach’s α and ICC single and average measures and standard error of measurement (SEM) reported in Watts and 95% confidence intervals (CI) for 0.025, 0.050, 0.075, and 0.100 kg·m^−2^ flywheel Romanian deadlift mean power outputs between complete sets over the 4 familiarization testing sessions for the internal and external attentional focus groups.

		Intraclass Correlation Coefficient (ICC) Internal Group					
Inertial Loads		Cronbach’s α	Single Measure	95% CI	Average Measure	95% CI	SEM
0.025 kg·m^2^ (Day 1–2)		0.76	0.61	0.26–0.82	0.76	0.41–0.90	41.86
0.025 kg·m^2^ (Day 2–3)		0.68	0.51	0.12–0.77	0.68	0.21–0.87	
0.025 kg·m^−2^ (Day 3–4)		0.83	0.71	0.41–0.87	0.83	0.58–0.93	

0.050 kg·m^2^ (Day 1–2)		0.59	0.43	0.01–0.73	0.61	0.02–0.84	49.41
0.050 kg·m^2^ (Day 2–3)		0.67	0.52	0.11–0.77	0.68	0.20–0.87	
0.050 kg·m^2^ (Day 3–4)		0.56	0.32	−0.06–0.64	0.48	−0.13–0.78	

0.075 kg·m^2^ (Day 1–2)		0.41	0.26	−0.19–0.62	0.41	−0.46–0.76	35.91
0.075 kg·m^2^ (Day 2–3)		0.77	0.57	0.18–0.80	0.73	0.31–0.89	
0.075 kg·m^2^ (Day 3–4)		0.47	0.26	−0.11–0.59	0.41	−0.25–0.74	

0.100 kg·m^2^ (Day 1–2)		−0.49	−0.20	−0.60–0.25	−0.51	−3.03–0.40	30.99
0.100 kg·m^2^ (Day 2–3)		0.68	0.30	−0.11–0.65	0.46	−0.24–0.78	
0.100 kg·m^2^ (Day 3–4)		0.66	0.48	0.09–0.75	0.65	0.17–0.85	
		**External Group**					
Inertial loads		Cronbach’s α	Single Measure	95% CI	Average Measure	95% CI	SEM
0.025 kg·m^2^ (Day 1–2)		0.79	0.66	0.33–0.85	0.80	0.49–0.92	37.06
0.025 kg·m^2^ (Day 2–3)		0.95	0.84	0.27–0.95	0.91	0.43–0.97	
0.025 kg·m^2^ (Day 3–4)		0.92	0.85	0.67–0.94	0.92	0.80–0.97	

0.050 kg·m^2^ (Day 1–2)		0.75	0.60	0.25–0.81	0.75	0.34–0.90	30.10
0.050 kg·m^2^ (Day 2–3)		0.85	0.64	0.14–0.85	0.78	0.25–0.92	
0.050 kg·m^2^ (Day 3–4)		0.94	0.89	0.76–0.96	0.94	0.86–0.98	

0.075 kg·m^2^ (Day 1–2)		0.65	0.50	0.08–0.76	0.66	0.15–0.87	26.25
0.075 kg·m^2^ (Day 2–3)		0.82	0.59	0.10–0.83	0.74	0.19–0.91	
0.075 kg·m^2^ (Day 3–4)		0.93	0.86	0.68–0.94	0.93	0.81–0.97	

0.100 kg·m^2^ (Day 1–2)		0.67	0.46	0.08–0.74	0.63	0.14–0.85	23.03
0.100 kg·m^2^ (Day 2–3)		0.79	0.55	0.08–0.81	0.71	0.14–0.89	
0.100 kg·m^2^ (Day 3–4)		0.93	0.87	0.71–0.95	0.93	0.83–0.97	

**Table 4 jfmk-09-00001-t004:** Cohen’s *d* effect size (ES), coefficient of variance (CV%), typical error (TE), and standard error of measurement (SEM) reported in Watts and smallest worthwhile change (SWC) at 0.5 for each inertial load in successive days for the internal and external attentional focus groups.

	Internal Group					
Inertial Loads	(ES)	CV (%)	TE	SWC0.5 (Watts)	SEM	Interpretation
0.025 kg·m^2^ (Day 1)	(D1–D2)- 0.15	7.50	67.53	14.48	36.10	Good
0.025 kg·m^2^ (Day 2)	(D2–D3)- 0.23	9.67	75.22	15.00	40.21	Good
0.025 kg·m^2^ (Day 3)	(D3–D4)- 0.12	10.09	86.88	18.96	46.44	Good
0.025 kg·m^2^ (Day 4)		5.58	73.05	10.28	39.04	Good
0.050 kg·m^2^ (Day 1)	(D1–D2)- 0.07	5.70	64.19	10.94	34.31	Good
0.050 kg·m^2^ (Day 2)	(D2–D3)- 0.05	14.45	110.84	15.44	59.25	Good
0.050 kg·m^2^ (Day 3)	(D3–D4)- 0.70	6.48	92.54	13.06	49.46	Good
0.050 kg·m^2^ (Day 4)		6.14	86.50	15.14	46.23	Good
0.075 kg·m^2^ (Day 1)	(D1–D2)- 0.21	6.51	49.14	18.76	26.26	Good
0.075 kg·m^2^ (Day 2)	(D2–D3)- 0.48	7.14	60.55	11.21	32.37	Good
0.075 kg·m^2^ (Day 3)	(D3–D4)- 0.69	5.38	79.51	17.18	42.50	Good
0.075 kg·m^2^ (Day 4)		7.87	66.58	14.81	35.59	Good
0.100 kg·m^2^ (Day 1)	(D1–D2)- 0.20	8.08	35.20	17.21	18.81	Good
0.100 kg·m^2^ (Day 2)	(D2–D3)- 1.22	8.21	47.70	11.41	25.50	Good
0.100 kg·m^2^ (Day 3)	(D3–D4)- 0.34	6.45	70.14	8.36	37.49	Good
0.100 kg·m^2^ (Day 4)		8.70	65.17	10.50	34.83	Good
	**External Group**					
Inertial loads	(ES)	CV (%)	TE	SWC0.5 (Watts)	SEM	Interpretation
0.025 kg·m^2^ (Day 1)	(D1–D2)- 0.22	7.03	79.02	13.35	42.24	Good
0.025 kg·m^2^ (Day 2)	(D2–D3)- 0.57	7.12	52.20	16.41	27.90	Good
0.025 kg·m^2^ (Day 3)	(D3–D4)- 0.11	9.59	64.97	19.45	34.73	Good
0.025 kg·m^2^ (Day 4)		4.61	80.70	11.70	43.14	Good
0.050 kg·m^2^ (Day 1)	(D1–D2)- 0.11	6.49	70.05	10.55	37.45	Good
0.050 kg·m^2^ (Day 2)	(D2–D3)- 0.42	8.45	47.49	24.86	25.39	Good
0.050 kg·m^2^ (Day 3)	(D3–D4)- 0.03	7.05	39.12	13.32	20.91	Good
0.050 kg·m^2^ (Day 4)		7.13	62.38	11.42	33.34	Good
0.075 kg·m^2^ (Day 1)	(D1–D2)- 0.30	10.67	61.53	11.24	32.89	Good
0.075 kg·m^2^ (Day 2)	(D2–D3)- 0.38	6.85	28.50	11.58	15.24	Good
0.075 kg·m^2^ (Day 3)	(D3–D4)- 0.20	9.47	33.78	10.88	18.06	Good
0.075 kg·m^2^ (Day 4)		7.43	62.78	14.91	33.56	Good
0.100 kg·m^2^ (Day 1)	(D1–D2)- 0.39	11.80	44.20	11.08	23.62	Good
0.100 kg·m^2^ (Day 2)	(D2–D3)- 0.65	7.81	23.55	12.10	12.59	Good
0.100 kg·m^2^ (Day 3)	(D3–D4)- 0.09	4.84	34.79	10.52	18.59	Good
0.100 kg·m^2^ (Day 4)		5.50	54.95	15.65	29.37	Good

**Table 5 jfmk-09-00001-t005:** Internal group (*n* = 7). Concentric power, eccentric power, and eccentric/concentric ratio by inertial load and testing day.

Variable	Day 1	Day 2	Day 3	Day 4
P_con_ 0.025 kg·m^2^	667.61 ± 172.66	665.67 ± 188.11	731.72 ± 155.30	724.72 ± 229.11
P_ecc_ 0.025 kg·m^2^	633.33 ± 153.57	667.33 ± 176.26	695.39 ± 154.36	685.56 ± 217.70
Ratio 0.025 kg·m^2^	0.98 ± 0.12	1.07 ± 0.16	1.08 ± 0.10	1.16 ± 0.11
P_con_ 0.050 kg·m^2^	601.83 ± 160.28	645.06 ± 148.90	655.44 ± 125.54	722.56 * ± 223.61
P_ecc_ 0.050 kg·m^2^	634.44 ± 152.66	695.06 ± 140.49	690.11 ± 136.99	765.56 * ± 237.55
Ratio 0.050 kg·m^2^	0.97 ± 0.12	1.02 ± 0.17	1.04 ± 0.13	1.06 ± 0.08
P_con_ 0.075 kg·m^2^	546.61 ± 149.01	564.11 ± 108.75	609.28 ± 128.83	674.33 * ± 234.24
P_ecc_ 0.075 kg·m^2^	601.44 ± 178.97	616.67 ± 95.67	643.28 ± 121.11	717.78 *# ± 267.82
Ratio 0.075 kg·m^2^	0.96 ± 0.07	1.06 ± 0.14	1.07 ± 0.16	1.13 ± 0.12
P_con_ 0.100 kg·m^2^	470.50 ± 100.13	492.44 ± 88.41	583.72 ± 131.67	618.67 ± 216.41
P_ecc_ 0.100 kg·m^2^	544.44 ± 115.96	548.72 ± 95.05	657.06 ± 149.79	657.72 ± 212.25
Ratio 0.100 kg·m^2^	0.95 ± 0.03	1.06 ± 0.10	1.06 ± 0.06	1.07 ± 0.06

P_con_ = peak concentric power output, P_ecc_ = peak eccentric power output, Ratio = eccentric overload ratio calculated by P_ecc_/P_con_ represented for each inertial load. * = significantly greater than day 1 (*p* ≤ 0.05); # = significantly greater than day 2 (*p* ≤ 0.05).

**Table 6 jfmk-09-00001-t006:** External group (*n* = 7). Concentric peak power, eccentric peak power, and eccentric/concentric ratio by inertial load and testing day.

Variable	Day 1	Day 2	Day 3	Day 4
P_con_ 0.025 kg·m^2^	598.61 ± 144.50	607.11 ± 274.84	631.06 ± 191.91	737.89 *# ± 242.12
P_ecc_ 0.025 kg·m^2^	525.56 ± 132.28	600.00 ± 348.76	637.83 * ± 251.22	720.06 *# ± 320.44
Ratio 0.025 kg·m^2^	0.90 ± 0.07	1.01 ± 0.11	1.09 ± 0.16	1.12 ± 0.16
P_con_ 0.050 kg·m^2^	619.56 ± 178.00	553.39 ± 150.90	610.17 ± 160.07	719.89 *# ± 201.83
P_ecc_ 0.050 kg·m^2^	633.00 ± 210.43	583.83 ± 225.85	678.06 # ± 304.86	826.83 *# ± 326.14
Ratio 0.050 kg·m^2^	0.95 ± 0.18	1.03 ± 0.22	1.08 ± 0.21	1.13 ± 0.16
P_con_ 0.075 kg·m^2^	539.89 ± 109.62	516.67 ± 111.98	578.50 ± 155.09	628.11 *# ± 213.66
P_ecc_ 0.075 kg·m^2^	588.67 ± 179.82	556.17 ± 174.47	642.67 # ± 260.92	739.33 *# ± 321.90
Ratio 0.075 kg·m^2^	1.00 ± 0.12	1.08 ± 0.22	1.08 ± 0.25	1.15 ± 0.20
P_con_ 0.100 kg·m^2^	473.89 ± 82.17	477.00 ± 90.39	501.89 ± 103.40	577.00 *# ± 176.50
P_ecc_ 0.100 kg·m^2^	535.22 ± 143.94	529.94 ± 157.45	575.83 ± 188.84	638.83 *# ± 248.19
Ratio 0.100 kg·m^2^	0.95 ± 0.11	1.12 ± 0.15	1.18 ± 0.18	1.12 ± 0.18

P_con_ = peak concentric power output; P_ecc_ = peak eccentric power output; ratio = eccentric overload ratio calculated with P_ecc_/P_con_, represented for each inertial load. * = significantly greater than day 1 (*p* ≤ 0.05); # = significantly greater than day 2 (*p* ≤ 0.05).

## Data Availability

The data presented in this study are available upon request from the corresponding authors.

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
