# Peer review of "Flywheel Romanian Deadlift: Intra- and Inter-Day Kinetic and Kinematic Reliability of Four Inertial Loads Using Cluster Sets"

_jfmk, 2023, doi:10.3390/jfmk9010001_

Round 1
Reviewer 1 Report
Comments and Suggestions for Authors
Abstract
The conclusion is not appropriately aligned with the stated objective. Please provide clearer emphasis on the discovered results in the abstract text.
Introduction
The introduction sufficiently lays out the background; however, the study's objective is not clearly understood. Are the measures referred to in the text the parameters of CS training (lines 90-91)? Will there be a measurement of cognitive function improvement for skill acquisition at any point? I understand that this is a hypothesis justifying potential results and should not be included in the study hypotheses related to the objective.
Methods
I suggest including an image of the FIT Romanian Deadlift (RDL) in the procedures, highlighting the initial and final positions of the movement from both frontal and lateral perspectives.
Please provide references for the selection of loads in FIT (lines 141-143). Additionally, include references for determining external and internal focus commands.
Results
The results were clearly presented.
Discussion
At the beginning of the discussion (lines 292-296), the study's objectives are presented differently from the abstract and introduction. In this section, the description of the objectives is clearer, and I believe it is more accurate. However, CS training is not mentioned. It is essential to better describe the study's objectives consistently throughout the text.
The limitation mentioned regarding the CS training interval underscores the importance of providing references in the methods that justify the selection of parameters for this protocol.
Conclusions
The information presented in lines 362 to 364 does not align with the study's objectives and methods. Therefore, it should not be included in the conclusions.
Author Response
Dear reviewers,
We would like to thank you all for your constructive and kind comments to assist in improving the manuscript. We also thank you all for kind complements provided. All these comments and complements were very much valued by the authors. We thank you for taking the time to review our work.
Please find the changes in bold and highlighted in yellow in the main manuscript. We have also provided the line numbers to assist in finding the changes made.
Thank you in advance for reviewing the changes we have made and believe the manuscript is much improved.
Kind regards
The author team
Reviewer 1
Abstract
The conclusion is not appropriately aligned with the stated objective. Please provide clearer emphasis on the discovered results in the abstract text.
Response:
Thank you for your comment. Please see lines 26-28.
Introduction
The introduction sufficiently lays out the background; however, the study's objective is not clearly understood. Are the measures referred to in the text the parameters of CS training (lines 90-91)?
Thank you for the comment. Please see lines 96-100.
Will there be a measurement of cognitive function improvement for skill acquisition at any point?
Response:
Thank you for the query. No. There was no cognitive function measurement.
I understand that this is a hypothesis justifying potential results and should not be included in the study hypotheses related to the objective.
Response:
Thank you for your comment. The hypothesis has been removed.
Methods
I suggest including an image of the FIT Romanian Deadlift (RDL) in the procedures, highlighting the initial and final positions of the movement from both frontal and lateral perspectives.
Thank you to the reviewer for the suggestion. We take note of the suggestion, however, the study currently has a total of 8 tables and figures. We believe the manuscript is long already with this number of tables and figures.
Please provide references for the selection of loads in FIT (lines 141-143). Additionally, include references for determining external and internal focus commands.
Response:
Thank you for the comment. The additional references have been added to lines 146-148 as per your recommendation for the selection of inertial loads.
For the attentional focus commands please see lines 166-167.
Results
The results were clearly presented.
Response: Thank you for the kind comment.
Discussion
At the beginning of the discussion (lines 292-296), the study's objectives are presented differently from the abstract and introduction. In this section, the description of the objectives is clearer, and I believe it is more accurate. However, CS training is not mentioned. It is essential to better describe the study's objectives consistently throughout the text.
Response:
Thank you for your comment. Please see lines 296-300.
The limitation mentioned regarding the CS training interval underscores the importance of providing references in the methods that justify the selection of parameters for this protocol.
Response:
Thank you for the comment relating to the reference of the chosen CS methodology. Please see the highlighted reference in the updated methods section at line 148.
Conclusions
The information presented in lines 362 to 364 does not align with the study's objectives and methods. Therefore, it should not be included in the conclusions.
Response:
Thank you for your comment. Lines 363 and 364 have been removed.
Reviewer 2 Report
Comments and Suggestions for Authors
General:
The study is in some ways simple and straightforward (not a bad thing), and in others complex (e.g., attentional focus). After reading the abstract, I was curious how the authors would fit everything together. Overall, the paper is well constructed, and the design is more interesting than most ‘reliability’ papers. The statistics are strong, and the writing is solid.
I have generally minor suggestions for improvement.
Title:
To make the title clearer up-front, I recommend the following tweak:
“Flywheel Romanian deadlift: Intra- and inter-day kinetic and kinematic reliability of four inertial loads using cluster sets”
Abstract:
Lines 13-14 are confusing. I think I understand what “In addition, to assessing the role of specific attentional focus instructions; internal and external cues.” Means, but it could be written far more elegantly.
The combination of attentional focus, different inertial loads, and biomechanical variables makes for interesting findings.
There are several relevant ‘effect sizes’. Thus, the authors should be sure to define what ‘ES’ is reported (Cohen’s d, Hedges’ g, Pearson’s r etc.).
Introduction:
I think that the relative portability of FIT should be mentioned as a reason for its growth in physical preparation and research.
While the introduction is already fairly long for a study like this, there is a distinct lack of mention of attentional focus. This can be addressed simply, and briefly, but presenting attentional focus as a variable to ‘prove/disprove’ that FIT Romanian deadlifts are reliable regardless of attentional focus.
Line 73: “A recent study by [9] reported…”
Lines 78-86: Seems like there is an extra space between the first zeros and the decimals? “0. 050” etc. This appears again in the results section (check throughout).
Methods:
The methods are generally clear, concise and well written. Only a few minor suggestions.
I thank the authors for using an effect size from an actual study for their sample size calculation! This is far better than using a generic 0.80 for unknown reasons. However, it would be nice to explain where the 0.60 came from. What is the context based on citation 22?
Line 145: Avoid abbreviations (e.g., ROM) that are not used more than a few times throughout the manuscript.
Great and robust statistical analysis.
Results:
The results section is excellent. The combination of words, tables, and figures work well to present all of the data that a reader could possibly want to make an informed decision regarding the reliability of FIT.
I have no suggestions for improvement.
Discussion:
The discussion is also strong, and appropriately long/detailed for this type of paper. I have only a few points for improvement.
While the authors mention attentional focus in the ‘Conclusions’, they should add 1-2 sentences in section 4 further highlighting the similarities/differences in reliability and learning curves between attentional foci. The attentional focus groups are really an extra layer of the study and deserves more attention in my opinion.
Author Response
Dear reviewers,
We would like to thank you all for your constructive and kind comments to assist in improving the manuscript. We also thank you all for kind complements provided. All these comments and complements were very much valued by the authors. We thank you for taking the time to review our work.
Please find the changes in bold and highlighted in yellow in the main manuscript. We have also provided the line numbers to assist in finding the changes made.
Thank you in advance for reviewing the changes we have made and believe the manuscript is much improved.
Kind regards
The author team
General:
The study is in some ways simple and straightforward (not a bad thing), and in others complex (e.g., attentional focus). After reading the abstract, I was curious how the authors would fit everything together. Overall, the paper is well constructed, and the design is more interesting than most ‘reliability’ papers. The statistics are strong, and the writing is solid.
I have generally minor suggestions for improvement.
Response:
Thank you for your positive feedback.
Title:
To make the title clearer up-front, I recommend the following tweak:
“Flywheel Romanian deadlift: Intra- and inter-day kinetic and kinematic reliability of four inertial loads using cluster sets”
Response:
Thank you for your suggestion. Please see the highlighted title in the updated document. We have applied your suggestion to make it clearer.
Abstract:
Lines 13-14 are confusing. I think I understand what “In addition, to assessing the role of specific attentional focus instructions; internal and external cues.” Means, but it could be written far more elegantly.
Response:
Thank you for the comment. Please see changes made in lines 11-15.
The combination of attentional focus, different inertial loads, and biomechanical variables makes for interesting findings.
Response:
Thank you for your kind comment.
There are several relevant ‘effect sizes’. Thus, the authors should be sure to define what ‘ES’ is reported (Cohen’s d, Hedges’ g, Pearson’s r etc.
Response:
Thank you for the comment. We have included Cohen’s d (Line 23).
Introduction:
I think that the relative portability of FIT should be mentioned as a reason for its growth in physical preparation and research.
Response:
Thank you for the comment. Please see lines 34-36.
While the introduction is already fairly long for a study like this, there is a distinct lack of mention of attentional focus. This can be addressed simply, and briefly, but presenting attentional focus as a variable to ‘prove/disprove’ that FIT Romanian deadlifts are reliable regardless of attentional focus.
Response:
Thank you for the comment. Please see lines 54-57.
Line 73: “A recent study by [9] reported.
Response:
Please see lines 79-81 for the change.
Lines 78-86: Seems like there is an extra space between the first zeros and the decimals? “0. 050” etc. This appears again in the results section (check throughout).
Response:
Thank you. Please see line 84, space removed.
Methods:
The methods are generally clear, concise and well written. Only a few minor suggestions.
Response:
Thank you for your kind words and feedback.
I thank the authors for using an effect size from an actual study for their sample size calculation! This is far better than using a generic 0.80 for unknown reasons. However, it would be nice to explain where the 0.60 came from. What is the context based on citation 22?
Response:
Thank you for the comment. The refence Tansel et al., 2008 “[22]” used an effect size of 0.6 in a study investigating hamstring eccentric training. This was used to guide the Gpower calculation prior to participant recruitment.
Line 145: Avoid abbreviations (e.g., ROM) that are not used more than a few times throughout the manuscript.
Response:
Thank you for the comment. The “ROM” abbreviation has been removed from the document as per recommendation.
Great and robust statistical analysis.
Response:
Thank you for the kind comment.
Results:
The results section is excellent. The combination of words, tables, and figures work well to present all of the data that a reader could possibly want to make an informed decision regarding the reliability of FIT.
I have no suggestions for improvement.
Response:
Thank you for the kind comments, we value them.
Discussion:
The discussion is also strong, and appropriately long/detailed for this type of paper. I have only a few points for improvement.
Response:
Thank you for the kind comment.
While the authors mention attentional focus in the ‘Conclusions’, they should add 1-2 sentences in section 4 further highlighting the similarities/differences in reliability and learning curves between attentional foci. The attentional focus groups are really an extra layer of the study and deserves more attention in my opinion.
Response:
Thank you for the comment. Please see lines 308-312.
Reviewer 3 Report
Comments and Suggestions for Authors
Dear Authors
As one of the reviewers, I express my personal scientific opinion on your work. I would like to reassure you that I was trying to be positive and constructive but particularly as fair and honest as possible to your work. The logical flow in the Introduction, the clear explanation provided in the Method’s section and the justification of the sample size using a-priori Power analysis are appreciated. I should also note that the originality of the study, the whole statistical approach used, the calculation of the Effect Size, the presentation of the ICC, the work done on tables and figures and the presentation of the limitations of the study are all positive points. The small sample size however, is a negative issue.
Please accept my judgment with a positive and constructive way.
Please see specific comments below:
Introduction:
1. Line 66: I would like to suggest: “A study by name et al… and at the end of the sentence you could place [17].
2. Line 73: same as above.
Methods:
3. Did you get any precaution concerning the diet of the athletes prior to each training session and/or official testing?
4. Based on your GPower sample size calculation, a total of 20 participants are required for securing statistical power. Although, this particular issue is discussed in your limitation’s section, how positive you are that with only n=14 (7 vs 7) the type I or II particularly statistical errors were not increased? What about the generalization of the results?
5. Based on the above, could you please consider including the words “A pilot study” in your title?
Author Response
Dear reviewers,
We would like to thank you all for your constructive and kind comments to assist in improving the manuscript. We also thank you all for kind complements provided. All these comments and complements were very much valued by the authors. We thank you for taking the time to review our work.
Please find the changes in bold and highlighted in yellow in the main manuscript. We have also provided the line numbers to assist in finding the changes made.
Thank you in advance for reviewing the changes we have made and believe the manuscript is much improved.
Kind regards
The author team
Dear Authors
As one of the reviewers, I express my personal scientific opinion on your work. I would like to reassure you that I was trying to be positive and constructive but particularly as fair and honest as possible to your work. The logical flow in the Introduction, the clear explanation provided in the Method’s section and the justification of the sample size using a-priori Power analysis are appreciated. I should also note that the originality of the study, the whole statistical approach used, the calculation of the Effect Size, the presentation of the ICC, the work done on tables and figures and the presentation of the limitations of the study are all positive points. The small sample size however, is a negative issue.
Please accept my judgment with a positive and constructive way.
Please see specific comments below:
Response:
Thank you to the reviewer for their kind words about our study. Thank you for taking the time to review in order to improve the manuscript.
Introduction:
- Line 66: I would like to suggest: “A study by name et al… and at the end of the sentence you could place [17]
Response:
Thank you for your comment. The reference has been amended as per recommendation. Please see lines 72.
- Line 73: same as above.
Response:
Thank you for your comment. Please see lines 79-81.
Methods:
- Did you get any precaution concerning the diet of the athletes prior to each training session and/or official testing?
Response:
Thank you for your comment. The participants were instructed to maintain their normal eating habits throughout the entirety of the research study.
- Based on your GPower sample size calculation, a total of 20 participants are required for securing statistical power. Although, this particular issue is discussed in your limitation’s section, how positive you are that with only n=14 (7 vs 7) the type I or II particularly statistical errors were not increased? What about the generalization of the results?
Response:
Thank you for the comment.
When using the effect size of 0.70 for the 0.05 kgm2 inertial load from day 2 to day 3 for the internal group, the power is equal to 0.86 for a post-hoc analysis suggesting that the final sample size of the current study was sufficient. Thus, this suggests that the risk of achieving type 1 or type 2 statistical errors were not increased. However, we do acknowledge as we have done in the limitations in the discussion that the sample size was reduced and unfortunately we were unable to recruit additional participants due to dropouts because of the impact of COVID-19.
- Based on the above, could you please consider including the words “A pilot study” in your title?
Response:
Thank you for your comment. We did consider including ‘A pilot study’ in the title based upon your suggestion. However, we decided against it based upon the article by Eldridge et al., (2016). As the current study was to examine the reliability of power measures when performing a flywheel RDL it did not meet the criterion of a pilot study as it was not looking at the feasibility of reliability of the particular exercise on a flywheel device.
Eldridge SM, Chan CL, Campbell MJ, Bond CM, Hopewell S, Thabane L, Lancaster GA; PAFS consensus group. CONSORT 2010 statement: extension to randomised pilot and feasibility trials. BMJ. 2016 Oct 24;355:i5239. doi: 10.1136/bmj.i5239. PMID: 27777223; PMCID: PMC5076380.